# Influenza H1 Mosaic Hemagglutinin Vaccine Induces Broad Immunity and Protection in Mice

**DOI:** 10.3390/vaccines7040195

**Published:** 2019-11-23

**Authors:** Brigette N. Corder, Brianna L. Bullard, Jennifer L. DeBeauchamp, Natalia A. Ilyushina, Richard J. Webby, Eric A. Weaver

**Affiliations:** 1School of Biological Sciences, Nebraska Center for Virology, University of Nebraska-Lincoln, Lincoln, NE 68503, USA; brigette.corder@huskers.unl.edu (B.N.C.); bbullard@huskers.unl.edu (B.L.B.); 2Department of Infectious Diseases, St. Jude Children’s Research Hospital, Memphis, TN 38105, USA; Jennifer.DeBeauchamp@stjude.org (J.L.D.); richard.webby@stjude.org (R.J.W.); 3Division of Biotechnology Review and Research II, Center for Drug Evaluation and Research, U.S. Food and Drug Administration, Silver Spring, MD 20993, USA; Natalia.Ilyushina@fda.hhs.gov

**Keywords:** influenza, universal, vaccine, mosaic, epitope-based vaccine, t-cell, cross-protection

## Abstract

Annually, influenza A virus (IAV) infects ~5–10% of adults and 20–30% of children worldwide. The primary resource to protect against infection is by vaccination. However, vaccination only induces strain-specific and transient immunity. Vaccine strategies that induce cross-protective immunity against the broad diversity of IAV are needed. Here we developed and tested a novel mosaic H1 HA immunogen. The mosaic immunogen was optimized in silico to include the most potential B and T cell epitopes (PBTE) across a diverse population of human H1 IAV. Phylogenetic analysis showed that the mosaic HA localizes towards the non-pandemic 2009 strains which encompasses the broadest diversity in the H1 IAV population. We compared the mosaic H1 immunogen to wild-type HA immunogens and the commercial inactivated influenza vaccine, Fluzone. When analyzed by ELISA, the mosaic immunogen induced stronger antibody responses against all four diverse H1 HA proteins. When analyzing T cell responses, again the mosaic immunogen induced stronger cellular immunity against all 4 diverse HA strains. Not only was the magnitude of T cell responses strongest in mosaic immunized mice, the number of epitopes recognized was also greater. The mosaic vaccinated mice showed strong cross-protection against challenges with three divergent IAV strains. These data show that the mosaic immunogen induces strong cross-protective immunity and should be investigated further as a universal influenza vaccine.

## 1. Introduction

Annually, influenza A virus (IAV) infects ~5–10% of adults and 20–30% of children worldwide, resulting in up to 650,000 deaths each year [1,2,3]. The ability of IAV to mutate, reassort, and infect multiple hosts results in a diverse viral population that causes annual epidemics and occasionally pandemic outbreaks. One of the most prominent examples is the 2009 influenza pandemic (Pdm09). A triple reassorted swine, avian, and human H1N1 influenza virus infected 24% of the global population [4,5]. Current influenza vaccine approaches are not well equipped to protect against novel strains as they induce a strain-specific immune response and matching vaccines can take up to six months to develop [6,7]. Over the past decade, licensed seasonal influenza vaccines have shown variable efficacy (10–60%) and limited cross-protection against drifted IAV strains [8,9]. Therefore, there is an urgent need for better vaccine strategies that will protect against the diverse influenza virus population.

Many universal influenza vaccine strategies have been investigated. Surface viral glycoproteins hemagglutinin (HA) and neuraminidase (NA) are the most common targets for vaccines, however, they provide limited cross-protection. Recent studies have targeted the HA stalk region which is more conserved than the HA head domain [10,11,12,13]. Consensus HA approaches have also been investigated to maximize coverage for IAV strains [14,15]. HA and NA approaches rely on the induction of antibody responses to provide protection against IAV. However, increasing evidence suggests that cellular immunity may be key to better vaccine cross-protection [16,17,18,19] and T cell depletion studies have shown that cellular immunity plays a significant role in protection against influenza [20,21]. Here, we investigate a mosaic HA immunogen which is computationally designed to induce strong cellular and humoral immunity.

In this study, a mosaic HA vaccine was designed to contain the most potential B and T cell epitopes (PBTE) across a spectrum of human H1 influenza viruses using the mosaic vaccine designer tool which predicts the sequence containing the most linear PBTE [22]. This mosaic strategy has shown promise for multiple pathogens including H5 influenza, Dengue, and HIV [23,24,25,26]. However, this is the first published study that applies the mosaic vaccine immunogen approach to human H1 influenza virus. This is important since the majority of circulating IAV strains are of the H1 and H3 subtypes. Here we analyzed the immune response and protection induced by a monovalent H1 HA mosaic immunogen expressed in an adenoviral vector.

## 2. Materials and Methods

### 2.1. Ethics Statement 

BALB/cJ mice aged 6–8 weeks were purchased from Jackson Laboratory (Bar Harbor, ME, USA). Mice were housed in the Life Sciences Annex building on the University of Nebraska—Lincoln (UNL) campus under the Association for Assessment and Accreditation of Laboratory Animal Care International (AAALAC) guidelines. The protocols were approved by the UNL Institutional Animal Care and Use Committee (IACUC) (Project ID 1217 and 1717: Influenza Vaccine Development). All animal experiments were carried out according to the provisions of the Animal Welfare Act, PHS Animal Welfare Policy, the principles of the NIH Guide for the Care and Use of Laboratory Animals, and the policies and procedures of UNL. Animal experiments at St. Jude Children’s Research Hospital (SJCRH) were approved by the SJCRH Animal Care and Use Committee in compliance with the National Institutes of Health and the Animal Welfare Act. All immunizations and bleeds were performed under either isoflurane or ketamine and xylazine-induced anesthesia.

### 2.2. Viruses

Influenza strains (A/New Caledonia/20/1999 NR-41799; A/Puerto Rico/8/1934 NR-348; A/Fort Monmouth/1/1947 NR-15568; A/California/07/2009 NR-13663; A/WS/33 NR-2759) were provided by Biodefense and Emerging Infections Research Resources Repository NIAID (BEI Resources, Manassas, VA, USA). Mouse-adapted viruses were obtained from ATCC (A/Puerto Rico/8/1934 (VR-95), A/Fort Monmouth/1/1947 (VR-1754)), and St Jude Children’s Research Hospital (A/Nanchang/1/1999). These viruses were passaged 4, 0, and 12 times, respectively, and the MLD_50_ was determined through serial dilution in BALB/c mice for lethal viruses. Viral stocks were grown in specific pathogen-free (SPF) research quality eggs provided by Charles River (Wilmington, MA, USA), inoculated on day 10 of incubation, and harvested after three days. Allantoic fluid was harvested from infected eggs and centrifuged at 200× *g* for 10 min. Aliquots of supernatant were stored at −80 °C. Viruses were quantified based on HAU and TCID_50_.

### 2.3. Mosaic Gene Design

The human influenza H1 mosaic (mosaic) hemagglutinin (HA) immunogen was designed using the Mosaic Vaccine Designer (Los Alamos National Laboratories Database, Los Alamos, U.S.A.). All unique full-length human H1 influenza HA sequences from 1918 to 2018 were downloaded from the Influenza Research Database (duplicate sequences and laboratory strains excluded). The resulting 6908 sequences were submitted to the Mosaic Vaccine Designer in fasta format with the following parameters: Cocktail Size: 3, Epitope Length: 9, Rare Threshold: 1, Run Time: 10 h, Population size: 200, Cycle Time: 10, Stall Time: 10, Internal Crossover Probability: 0.5. The first mosaic immunogen was used for this study.

### 2.4. Phylogenetic and Sequence Analysis of Mosaic Gene

The 6908 unique H1 HA sequences used for the mosaic immunogen design were aligned with ClustalX2.1 [27] with fast-approximate pairwise parameters: gap penalty: 3; k-tuple size: 1; top diagonals: 5; window size: 5. The output nexus file was used in PAUP version 4.0a165 [28] to design a neighbor joining tree. The mosaic and relevant HA strains are labelled on the phylogenetic tree. The percent similarity using Blosum62 with threshold 1 and percent identity between the HA protein sequences was calculated using Geneious 11.0.5. 

### 2.5. Structural Analysis of Mosaic Gene

The mosaic, A/PR/8/34, and A/TX/05/09 (Pdm09) HA amino acid sequences were submitted to the SWISS-MODEL web server (Basel, Switzerland) [29]. The models with the highest Global Model Quality Estimation (GMQE), ranging from 0 to 1, and Qualitative Model Energy Analysis (QMEAN) scores below 4 were selected as the most reliable structures [30]. The mosaic, A/PR/8/34, and Pdm09 models used template 6n41.1.B with GMQE values of 0.81, 0.82, 0.79, and QMEAN values of −0.39, 0.25, −0.29, respectively. All models were uploaded to PyMOL (PyMOL Molecular Graphics System, version 2.3.2, Schrodinger LLC, New York, NY) for further visualization.

### 2.6. Recombinant Adenovirus Type 5 Plasmid Construction

The mosaic, A/Puerto Rico/8/1934 (NCBI Reference Sequence: NP_040980.1), and A/Texas/05/09 (Pdm09) (GenBank: ACU13094.1) HA genes were codon-optimized for human gene expression. Each DNA fragment was synthesized by GenScript (Piscataway, NJ, USA) and cloned into pcDNA3.1 mammalian expression vector with directional restriction enzymes sites for downstream cloning. The AdEasy Adenoviral Vector System (Agilent, Santa Clara, CA, USA) was used to make recombinant Adenovirus 5 (Ad5) lacking the E1 and E3 genes. The HA genes were cloned into the pShuttle-CMV vector from the AdEasy kit using T4 DNA ligase (NEB, Ipswitch, MA, USA). The plasmids were linearized and transformed into BJ5183 electrocompetent cells along with the pAdEasy-1 vector (Adenovirus type 5) for homologous recombination. During recombination, the HA gene is inserted into the E1 region of the Ad5 genome. Recombinants were confirmed by restriction digest and sequenced prior to transformation into XL1 cells for midiprep with the Qiagen Plasmid Midi Kit (Qiagen, Germantown, MD, USA).

### 2.7. Recombinant Adenovirus Rescue, Purification, and Quantification

The recombinant Ad5 genomes with HA inserts (Ad5-Mosaic; Ad5-A/PR/8/34; Ad5-Pdm09) were linearized and buffer exchanged with Strataprep PCR purification kit (Agilent, Santa Clara, CA, USA). Polyfect transfection reagent was used to transfect this linearized DNA into 293 cells. After cytopathic effects (CPE) and plaque formation was observed, cells were harvested and subjected to 3 freeze-thaw cycles to release virus. Recombinant virus was amplified through passages in 293 cells up to a Corning 10-cell stack flask (~6300 cm^2^). Virus was purified with 2 consecutive CsCl gradients and desalted though Econo-Pac 10DG Desalting Columns (Bio-Rad, Hercules, CA, USA) before storage at −80 °C in Ad-tris buffer (100 mM NaCl, 20 mM Tris-HCl, 1 mM MgCl_2_^•^6H_2_O, 10% glycerol). The virus particle quantity was measured with a NanoDrop Lite spectrophotometer (Thermo Fisher, Waltham, MA, USA) at OD260. 

### 2.8. Western Blotting

HA protein expression from recombinant Adenovirus 5 (rAd5) was confirmed by Western blot. Confluent 293 cells were infected with 500 virus particles (vp) per cell of rAd5 and incubated at 37 °C and 5% CO_2_ for 48 h prior to harvest. Laemmli buffer plus 2-mercaptoethanol was used to denature cells before inactivation at 100 °C for 10 min. The samples were loaded onto a 12.5% SDS-PAGE gel and separated by electrophoresis before transfer onto a nitrocellulose membrane. The membrane was blocked with 5% non-fat powdered milk in tris-buffered saline and Tween 20 (TBST) for 30 min before incubation in goat polyclonal anti-influenza virus H1 (H0) hemagglutinin (HA) A/PR/8/34 antibody (NR-3148; BEI resources, Manassas, VA, USA) at 1:2000 and mouse anti-GAPDH (sc-47724; Santa Cruz Biotechnology, Inc., Dallas, TX, USA) at 1:2000 in TBST 1% milk overnight at 4 °C. The membrane was washed 3× in TBST before incubation with goat anti-mouse-HRP conjugated antibody (Millipore Sigma, Burlington, MA, USA) at 1:2000 and goat IgG HRP-conjugated antibody (HAF109; R&D Systems, Minneapolis, MN, USA) at 1:2000 in TBST 1% milk for one hour at room temperature (RT). After three washes with TBST, membrane was developed with SuperSignal West Pico Chemiluminescent Substrate (Thermo Scientific, Waltham, MA, USA). 

### 2.9. Tissues for Humoral and Cellular Assays

Female BALB/cJ mice were immunized with 10^10^ vp of indicated recombinant adenovirus 5 (rAd5), 600 ng of Fluzone (approximately 30× a human dose), or control phosphate buffered saline (PBS). Mice were immunized with a prime or prime-boost of the same dose three weeks after initial vaccination. All immunizations were performed intramuscularly using a 29.5-gauge needle to administer 25 µL in both quadriceps. At three weeks post-prime or eight days post-boost, animals were terminally bled via cardiac puncture and spleens were harvested. Blood was centrifuged at 6000× *g* for two minutes in BD microtainer blood collection tubes (Becton Dickinson, Franklin Lakes, NJ, USA) to collect sera which was used for ELISA and hemagglutination inhibition (HAI) assays. Spleens were passed through a 40 µm nylon cell strainer (Thermo Fisher, Waltham, MA, USA), red blood cells lysed using ACK lysis buffer (150 mM NH_4_Cl, 10 mM KHCO_3_, 0.1 mM Na_2_EDTA), and spleenocytes were resuspended in RPMI with 5% FBS before analysis by ELISpot assay.

### 2.10. Hemagglutination Inhibition Titers

Serum samples were treated with a 1:3 dilution of serum: receptor destroying enzyme (370013; Denka Seiken, Tokyo, Japan) at 37 °C overnight, heat inactivated at 56 °C for 30 min, and tested by hemagglutination inhibition (HAI) assay with 0.5% chicken red blood cells. Briefly, serum was serial diluted two-fold in a 96 well V-bottom plate starting at a 1:10 dilution. An equal volume (25 uL) of 4 HAU dilution of virus was added to each well and incubated for 30 min before adding 50 μL of 0.5% chicken red blood cells and incubating 30 min before reading hemagglutination pattern.

### 2.11. ELISA

Immunolon 4 HBX microtiter 96-well strips (Thermo Fisher, Waltham, MA, USA) were coated with 200 ng/well of recombinant HA protein from BEI Resources, Manassas, VA, USA NIAID (A/California/04/2009, NR-15749; A/PR/8/34, NR-19240; A/Brisbane/59/07, NR-28607; A/New Caledonia/20/99, NR-48873) in bicarbonate/carbonate buffer overnight at 4 °C. Plates were blocked with 2% bovine serum albumin (BSA) in PBS and incubated for two hours at RT. Sera was serial diluted two-fold in PBS with 1% BSA starting with a 1:100 dilution in 100 μL per well. After two hours of incubation at RT, plates were washed 4× with phosphate buffered saline with Tween 20 (PBST) and 2× with PBS before incubation with 1:5000 goat anti-mouse-HRP antibody (sc-47724; Santa Cruz Biotechnology, Inc., Dallas, TX, USA) in 1% BSA at RT for one hour. Plates were washed 4× with PBST and 2× with PBS prior to development with 1-Step Ultra TMB-ELISA (Thermo Fisher, Waltham, MA, USA). Development was stopped with the addition of 2M sulfuric acid. The OD450 was measured using SpectraMax i3x Multi-Mode microplate reader (Molecular Devices, San Jose, CA, USA). Responses ≥2 times the negative control absorbance were considered positive and used to calculate the geometric mean titer (GMT). 

### 2.12. ELISpot Assay

The cellular responses to vaccination were measure by IFN-γ ELISpot assay with peptide arrays (A/New Caledonia/20/1999 NR-2606; A/Puerto Rico/8/1934 NR-18973; A/California/07/2009 NR-19244; A/Brisbane/59/2007 NR-18970) provided by Biodefense and Emerging Infections Research Resources Repository NIAID (BEI Resources, Manassas, VA, USA). All peptide arrays span the entire HA gene and consist of 13 to 17-mers with 11 or 12 amino acid overlap. Potential immunogenic peptides were identified using a matrix of peptide pools and confirmed using individual peptides, whereas total T cell responses were performed with a pool of all peptides in the array. Polyvinylidene difluoride-backed 96-well plates (MultiScreen-IP, Millipore Sigma, Burlington, MA, USA) were coated with 50 µL of anti-mouse IFN-γ mAb AN18 (5 µg/mL; Mabtech, Stockholm, Sweden) overnight at 4 °C. Plates were washed 4× with PBS and blocked in RPMI with 5% FBS at 37 °C for one hour. Equal volumes (50 µL) of the single-cell suspension splenocytes (2 × 10^6^ cells/mL) and peptide (5 ug/mL) were added to the wells before overnight incubation at 37 °C and 5% CO_2_. Plates were washed 6× with PBS and incubated with 100 µL of biotinylated anti-mouse IFN-γ R4-6A2 mAb (1:1000 dilution; Mabtech, Stockholm, Sweden) diluted in PBS with 1.0% FBS. After an hour incubation at RT, Plates were washed 6× with PBS and incubated with 100 µL of streptavidin-alkaline phosphatase conjugate (1:1000 dilution; Mabtech, Stockholm, Sweden) diluted in PBS 1.0% FBS for one hour. The plates were washed 6× with PBS before development. 100 µL of BCIP/NBT (Plus) alkaline phosphatase substrate (Thermo Fisher, Waltham, MA, USA) was added to each well and development was stopped by rinsing several times in dH_2_O. Plates were air dried prior to analysis by an automated ELISpot plate reader (AID iSpot Reader Spectrum; Autoimmun Diagnostika GmbH, Straberg, Germany). Data is shown as spot-forming cells (SFC) per 10^6^ splenocytes.

### 2.13. Mouse Challenge Studies

Groups of five female BALB/c mice (~8 weeks old) were immunized intramuscularly at day 0 with either 10^10^ vp or 10^8^ vp of Ad5-Mosaic, Ad5-A/PR/8/34, Ad5-Pdm09, control PBS, or 600 ng of HA (approximately 30× a human dose) from the 2018–2019 Fluzone vaccine generously donated from Bryan Health, Lincoln, NE, USA. At three weeks post-immunization, all mice were challenged intranasally with 100 MLD_50_ of A/PR/8/34 and A/FM/1/47 or 10^3.6^ TCID_50_ equivalents of A/Nanchang/1/99. All mice were monitored daily for weight loss and survival. Mice were humanely euthanized at 25% weight loss.

### 2.14. Statistical Analysis

All data were analyzed using GraphPad Prism 8.2 software (GraphPad, San Diego, CA, USA). Data are expressed as the mean or geometric mean titer (GMT) with standard error (SEM). Survival curves were analyzed using the log rank test while weight loss, HAI, ELISA, and ELISpot data was analyzed by one-way ANOVA with Bonferroni multiple comparison test. A *p*-value < 0.05 was considered statistically significant (* *p* < 0.05; ** *p* < 0.01; *** *p* < 0.001; **** *p* < 0.0001).

## 3. Results

### 3.1. Design and Characteristics of the Human H1 HA Mosaic Immunogen

We sought to design and test a mosaic hemagglutinin (HA) immunogen representing as much of the H1 influenza virus population as possible. Briefly, we submitted all 6908 unique full-length human H1 HA to the Mosaic Vaccine Designer server in order to predict an immunogen with the most PBTE [31]. The resulting full-length human H1 mosaic (mosaic) HA was analyzed based on its phylogeny and structure (Figure 1). When compared to all unique human H1 HA sequences, the mosaic sequence clustered with pre 2009 seasonal influenza viruses on the phylogenetic tree (Figure 1A). Strains used throughout this study were compared based on amino acid percent identity and percent similarity (Figure 1B). The mosaic HA is closely related to A/Brisbane/59/07 and least genetically similar to the pandemic 2009 strains.

Since the mosaic HA sequence was generated de novo, there was concern for the structure and folding of the encoded protein. To help alleviate these concerns we submitted the mosaic protein sequence to the SWISS-MODEL web server to predict its structure (Figure 1C). In addition to the mosaic gene, the wild-type HA genes from A/PR/8/34 and A/Texas/05/09 pdm09 were submitted. The SWISS-MODEL server uses homology modelling between the submitted sequence and a similar template sequence with known structure to predict folding [30]. The mosaic HA was predicted to fold into a conformation similar to the template and wild-type HA proteins. 

Based on the support of the SWISS-MODEL analyses we next wanted to test the expression of the mosaic HA. The mosaic gene was cloned into a replication defective Adenovirus type 5 (Ad5) viral vector and subsequent protein expression was confirmed by Western blot. As controls we compared the mosaic HA expression with that of two wild-type HA genes expressed in an Ad5 vector, A/Texas/05/09 (Ad5-Pdm09) and A/Puerto Rico/8/34 (Ad5-A/PR/8/34) (Figure 1D). The mosaic HA was detected at similar levels to the wild-type comparators, Appendix A.

### 3.2. Vaccine-Induced Antibody Responses

Mice were immunized with a prime or prime-boost strategy of 10^10^ virus particles (vp) of recombinant Adenovirus per immunization. Sera from vaccinated mice three weeks post-prime or eight days post-boost was analyzed by hemagglutination inhibition (HAI) assays using various H1 influenza strains (Figure 2). Mice vaccinated with Ad5-mosaic showed a strong antibody response against the genetically similar A/NC/20/99 strain (Figure 2A). When boosted, vaccinated mice also induced antibodies against the mismatched A/WS/33 strain (Figure 2B). Mice immunized with recombinant Ad5 expressing wild-type HAs were able to induce robust antibody responses against matched IAV strains. Both the Ad5-Pdm09 and Fluzone vaccines include a Pdm09 immunogen and displayed antibodies against the matched A/CA/07/09 (Pdm09) strain (Figure 2E). The Ad5-A/PR/8/34 vaccine induced antibodies against the matched A/PR/8/34 strain and the similar A/WS/33 strain. However, none of the wild-type HA or Fluzone showed detectable antibodies against mismatched strains.

Previous studies have suggested that enzyme linked immunosorbent assays (ELISA) are more sensitive than HAI assays to detect antibodies against HA, especially those against the HA stalk region [32,33]. To further characterize the humoral response to vaccination we also analyzed the sera from immunized mice by ELISA (Figure 3). As expected, mice vaccinated with Adenovirus expressing wild-type HA were able to induce high antibody titers against matched strains (Figure 3A,B). However, the wild-type immunogens were unable to produce high antibody titers when mismatched recombinant HA proteins were used (Figure 3C,D). Importantly, the Ad5-mosaic vaccine was able to induce significant antibody titers against all four divergent influenza strains, including the distant A/California/04/07 (Pdm09) strain. Indeed, the mosaic HA induced higher antibody titers than all other groups except when the wild-type HA vaccines matched the influenza strain. When all vaccines and HA strains were mismatched, the Ad5-mosaic vaccinated mice showed significantly higher antibody titers than other vaccine groups (Figure 3C,D).

### 3.3. Vaccine Induced T Cell Responses

The mosaic vaccine was designed to maximize the PBTE across the viral population, thus biasing the immune response towards greater cross-protection. To determine if this was the case, cellular responses from vaccinated mice were measured by IFN-γ ELISpot assay (Figure 4). Overlapping peptide libraries spanning the entire HA gene for each strain were used for the ELISpot assays. Mice vaccinated with Adenovirus expressing a wild-type HA were able to induce strong T cell responses against matched strains (Figure 4A,B). However, the Ad5-mosaic vaccinated mice were able to generate strong T cell responses across all four divergent influenza strains. Encouragingly, Ad5-mosaic vaccinated mice were able to produce similar levels of T cells against the A/PR/8/34 strain as did those immunized with the matched Ad5-A/PR/8/34 vaccine. These data suggest that the mosaic immunogen induces strong cross-reactive T cell responses.

The breadth of the cellular response to vaccination was analyzed by epitope mapping. All immunostimulatory peptides identified from A/Brisbane/59/07, A/New Caledonia/20/99, and A/California/07/09 (Pdm09) HA were mapped using IFN-γ ELISpot assays (Figure 5). All peptides and their relative location in the HA gene are shown for the mosaic (green), A/PR/8/34 (dark blue), and Pdm09 (light blue) vaccines (Figure 5A). All vaccines induced cellular immunity against two HA2 epitopes that were broadly cross reactive for all three HA peptide pools. The Pdm09 vaccine induced cellular responses against additional epitopes in the matching HA1. The mosaic vaccine induced responses against HA1 epitopes in all three peptide pools. 

Peptide mapping is an important indicator of T cell immunity, but there are many other factors involved in cellular responses including proteasome cleavage, antigen processing and presentation, and major histocompatibility complex (MHC) type [21]. To further investigate the biological relevance of the immunostimulatory peptides identified in Figure 5, sequences of positive peptides were submitted to the Immune Epitope Database and Analysis Resource. This database contains published reports of epitopes studied in multiple hosts. Many of the peptides identified by IFN-γ ELISpot were reported to bind multiple MHC complexes, qualitatively bind B cells, induce cytokines, stimulate T cell cytotoxicity, as well as induce other immune responses (Appendix A). Although the database does not have an exhaustive list of epitopes, the immunostimulatory peptides reported here could have additional biological relevance not yet reported.

### 3.4. Protection Against Divergent Influenza Virus Challenges 

The gold standard for vaccine efficacy is protection against challenge. Mice immunized with a single dose of 10^10^ or 10^8^ vp of Ad5-mosaic, Ad5-A/PR/8/34, Ad5-Pdm09, Fluzone, or PBS were challenged with sub-lethal or 100MLD_50_ of mouse-adapted IAV. Challenged mice were monitored daily for weight loss and humanely sacrificed if more than 25% initial body weight was lost.

#### 3.4.1. Influenza Virus A/Nanchang/1/99 challenge

All mice were challenged intranasally three weeks after vaccination with a sublethal dose of A/Nanchang/1/99. All Adenovirus vectored vaccines provided strong protection against weight loss for both vaccination doses (Figure 6). Even the divergent Ad5-Pdm09 vaccine showed minimal weight loss for this sub lethal A/Nanchang/1/99 challenge. In contrast, mice immunized with whole inactivated Fluzone or PBS displayed severe weight loss. Interestingly, the A/TX/05/09 HA included in the Ad5-Pdm09 vaccine is almost identical (99.47% similar) to the A/Michigan/45/15 HA included in the whole inactivated Fluzone vaccine, but only the Ad5-Pdm09 vaccine was protective against weight loss. This suggests that gene delivery and/or dose of immunogens can impact vaccine cross-protection. 

#### 3.4.2. Influenza Virus A/Fort Monmouth/1/47 challenge

To further investigate the protection of Ad5-mosaic, vaccinated mice were challenged intranasally with a lethal dose of A/FM/1/47 virus. Mice vaccinated with 10^10^ vp Ad5-mosaic were completely protected against weight loss and death (Figure 7). Even at the lower vaccination dose, Ad5-mosaic vaccinated mice showed 100% survival and minor weight loss. The A/PR/8/34 HA included in the Ad5-A/PR/8/34 vaccine is closely related (94.35% similar) to the A/FM/1/47 challenge strain. Mice vaccinated with the either dose of Ad5-A/PR/8/34 showed minor weight loss and 100% survival. In contrast, mice vaccinated with a distant Pdm09 immunogen (Ad5-Pdm09 or Fluzone) showed increased weight loss and death for the lethal A/FM/1/47 virus challenge. These data show that the mosaic vaccine was able to protect mice against the A/FM/1/47 strain at comparable levels to the similar A/PR/8/34 immunogen. Furthermore, the mosaic vaccine provided better protection than mismatched Pdm09 immunogens or Fluzone.

#### 3.4.3. Influenza Virus A/Puerto Rico/8/34 challenge

Vaccinated mice were challenged at three weeks post-immunization with lethal A/PR/8/34 virus. As expected, mice immunized with either dose of the matched Ad5-A/PR/8/34 vaccine showed minor weight loss and 100% survival against the A/PR/8/34 challenge (Figure 8). All but one mouse immunized with 10^8^ vp of Ad5-mosaic showed minimal weight loss and 100% survival. This outlier could be due to individual variation between mice or suboptimal vaccination. However, mice vaccinated with the high dose of Ad5-mosaic showed minor weight loss and 100% survival. Both the adenovirus (Ad5-Pdm09) and inactivated (Fluzone) Pdm09 vaccines did not provide protection against weight loss and death for this divergent lethal challenge. Overall, the Ad5-mosaic vaccine was able to protect mice from the lethal A/PR/8/34 challenge. 

## 4. Discussion

Mosaic immunogens have been explored over the past decade as a vaccination strategy to protect against viruses with high population diversity, such as influenza H5 and HIV [24,25,34]. By maximizing the PBTE coverage across the population, mosaic immunogens are designed to induce stronger immune responses [22]. Here we designed a mosaic HA immunogen for human H1 influenza. The synthetic mosaic protein localizes towards the pre 2009 H1 population due to the higher degree of diversity in these viruses relative to the post 2009 pandemic viruses. This is important since the mosaic immunogen is designed to provide the greatest level of protection against the most diverse strains. Clearly, the greatest level of genetic diversity exists outside of the Pdm09 cluster. The mosaic HA was cloned into an Adenovirus type 5 vector and used to vaccinate mice in order to evaluate the cross-protective immune responses. Mice immunized with Ad5-mosaic induced high antibodies across the diverse H1 population when measured by ELISA, but minimal responses when measured by HAI assay. This difference could be due to the sensitivity of each assay or due to the ability of ELISA to detect a broader range of HA antibodies, especially antibodies to the HA stalk region [32,33]. 

The mosaic vaccine induced strong cellular responses across diverse influenza strains. Previous studies show that inactivated vaccines induce much weaker cellular responses than other vaccine strategies [35,36]. As expected, the Fluzone vaccinated mice elicited low T cell responses to all four influenza HA virus strains. In contrast, Ad5-vectored vaccines induced robust T cell responses with the Ad5-mosaic vaccine generally inducing stronger responses across the breadth of tested HAs with responses focused on two areas of HA1 and HA2 (Figure 5). Since epitope mapping is specific to the MHC complex of BALB/c mice, we wanted to explore whether these epitopes have been previously linked to immune responses in other models. Immunostimulatory peptide sequences were submitted to the Immune Epitope Database and Analysis Resource to identify additional published immunological information (Appendix A). The results show that the immunostimulatory peptides identified in Figure 5 have been reported for multiple mouse, human, and non-human primate MHC types. In addition, most of the immunostimulatory peptides have been linked to B and T cell immune responses. Importantly, the database does not reflect a complete report for each peptide and additional biological relevance may be discovered in future studies. 

Using a sublethal A/Nanchang/1/99 challenge we were able to show greater cross-protection from the adenoviral vectored vaccines than the inactivated Fluzone vaccine (Figure 6). All mice immunized with adenovirus expressing HA, including the divergent Ad5-Pdm09 vaccinated mice, were protected from weight loss. In contrast, the Fluzone vaccine, which contains a similar HA to the Ad5-Pdm09 vaccine, was unable to provide protection (weight loss in this group was comparable to the PBS negative control). The stronger cross-protection from adenovirus vectored vaccines than inactivated virus vaccines has been previously shown for IAV [37]. In our study, the reduced effectiveness of the inactivated vaccine corresponded to lower antibody and overall T cell responses. Despite increased immunogenicity, adenoviral vectored vaccines are currently not licensed for human vaccine use. Therefore, additional vaccine platforms could be investigated to maximize the immune response to vaccination. 

Mice immunized with Ad5-mosaic showed protection against both highly lethal influenza challenges conducted. The A/PR/8/34 and A/FM/1/47 challenge viruses are both similar to the A/PR/8/34 HA included in the Ad5-A/PR/8/34 vaccine. For both 100MLD_50_ challenges, Ad5-mosaic vaccinated mice were protected at comparable levels to the matched Ad5-A/PR/8/34 vaccinated animals. Importantly, Ad5-mosaic vaccinated animals only showed detectable antibodies against A/PR/8/34 and A/FM/1/47 when measured by ELISA, but no detectable antibodies when measured by HAI assay. In contrast, the mismatched Ad5-Pdm09 vaccinated mice showed severe weight loss and death for both the A/PR/8/34 and A/FM/1/47 challenges. 

In addition to being superior to the inactivated vaccine at inducing antibody responses, another benefit of adenovirus vectored vaccines is their ability to induce T cell responses. We detected T cell responses in all adenovirus immunized animals with the Ad5-mosaic vaccine able to induce robust responses to each of the H1 HAs measured. While we determined cross-reactive immune responses only to H1 antigens, it is likely that protection against other influenza viruses will require multiple immunogens. Co-administration of mosaic immunogens representing different influenza HA subtypes including H2, H3, and H5, as well as neuraminidase types N1 and N2, may have promise as universal influenza vaccine immunogens. Previous work has shown that multiple synthetic immunogens can be combined into a multivalent vaccine without losing efficacy of the individual components [15]. In this study we show that our mosaic immunogen is able to elicit broad levels of humoral and cellular immunity and cross-protection against divergent H1N1 influenza virus strains. Based on these data we believe that this strategy should be further explored as a strategy for the development of universal influenza vaccine immunogens.

## 5. Conclusions

This is the first study characterizing a mosaic H1 hemagglutinin as a broadly-reactive immunogen for the prevention of H1N1 influenza virus infections. Our mosaic design is even more unique than other published strategies because we used only the unique full-length H1 HA sequences in the database to create the mosaic gene. This excluded repeated sequences that would have biased the immunogen and reduced the overall coverage of protection. The mosaic HA immunogen was expressed in an Adenovirus type 5 replication-defective virus. The H1 mosaic immunogen induced strong cellular and humoral immune responses against divergent H1N1 influenza virus strains. In addition, the H1 mosaic immunogen also provided complete protection against a sublethal infection with the A/Nanchang/1/1999 influenza virus. More impressively, the mosaic H1 immunogen provided complete protection against death when challenged with lethal doses of A/Fort Monmouth/1/47 and A/Puerto Rico/8/34 influenza viruses. Our data indicate that mosaic immunogens can induce broadly-reactive immunity and warrant further investigation as a universal influenza vaccine strategy. 

## 6. Patents

“Novel Synthetic Influenza Antigens for us as Universal Vaccines.” Disclosed to the University of Nebraska, Lincoln, NU Tech Ventures. Provisional Patent Application #62734791, filed 09/21/2018.

## Figures and Tables

**Figure 1 vaccines-07-00195-f001:**
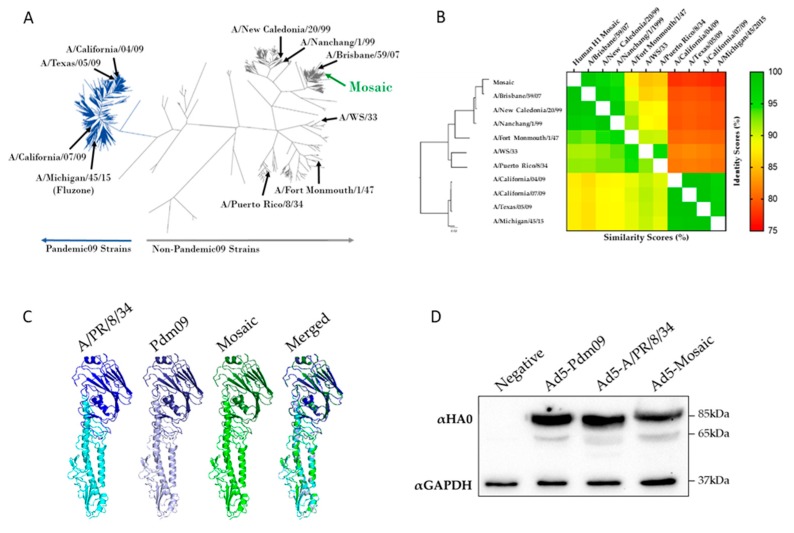
Structural and sequence analysis of the mosaic immunogen. (**A**) All unique human H1 HA protein sequences were used to make a neighbor-joining phylogenetic tree. Representative strains are labelled and pandemic 2009 strains are shown in dark blue. (**B**) HA proteins of strains used throughout this study were analyzed based on phylogeny, percent identity (upper right), and percent similarity (lower left). (**C**) A/PR/8/34, A/TX/05/09 (Pdm09) and Mosaic HA sequences were submitted to the SWISS-MODEL server and visualized with PyMOL software to predict the protein structure. The HA head (dark) and stalk (light) regions are shown along with a merged image of all three HA immunogens. (**D**) A/PR/8/34, Pdm09, and the mosaic HA were expressed in recombinant Adenovirus type 5 (rAd5). Cells were infected with 500 virus particles per cell of rAd5 and HA expression was detected by western blot. The western blot detected the HA0 (85kDa) and HA1 (65kDa) bands at similar levels for all rAd5.

**Figure 2 vaccines-07-00195-f002:**
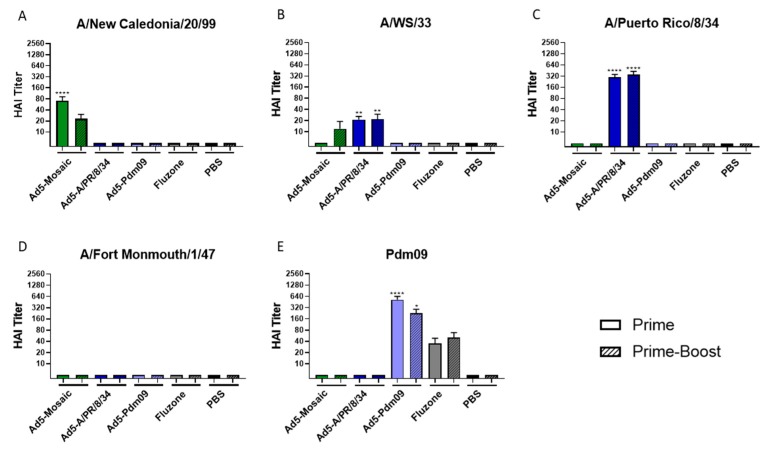
HAI titers after vaccination. Mice were immunized with Ad5-Mosaic, Ad5-A/PR/8/34, Ad5-Pdm09, Fluzone, or PBS using a prime (*n* = 8) or prime/boost (*n* = 5) strategy. Sera was analyzed by HAI assay to measure antibodies against influenza strains (**A**) A/New Caledonia/20/99, (**B**) A/WS/33, (**C**) A/Puerto Rico/8/34, (**D**) A/Fort Monmouth/1/47, and (**E**) A/California/07/09 (Pdm09). Data is expressed as the mean with standard error (Data was analyzed with one-way ANOVA and Bonferroni multiple comparison and compared to PBS; **** *p* < 0.0001, ** *p* < 0.01, * *p* < 0.05).

**Figure 3 vaccines-07-00195-f003:**
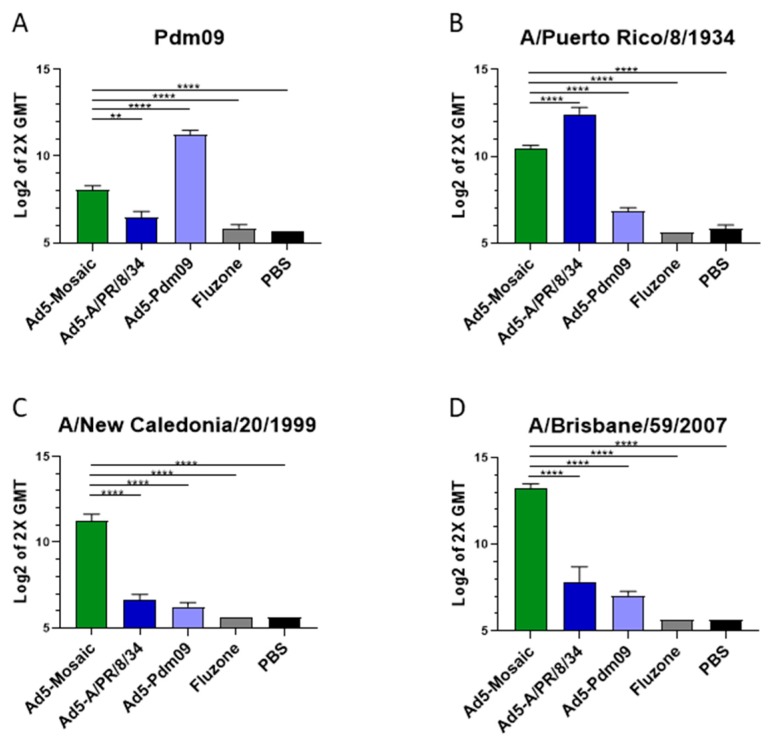
Antibody response after single-shot vaccination. Mice were immunized with 10^10^ virus particles of recombinant Ad5-Mosaic, Ad5-A/PR/8/34, Ad5-Pdm09, Fluzone, or PBS and sera harvested three weeks later (*n* = 5). ELISA were performed with recombinant HA protein from influenza strains (**A**) A/CA/07/09 (Pdm09), (**B**) A/PR/8/34, (**C**) A/NC/20/99, and (**D**) A/Brisbane/59/07. Responses greater than two times the geometric mean titer (GMT) were considered positive. Data is shown as the Log2 mean with standard error (data was analyzed with one-way ANOVA and Bonferroni multiple comparison and compared to the Ad5-mosaic; **** *p* < 0.0001, ** *p* < 0.01).

**Figure 4 vaccines-07-00195-f004:**
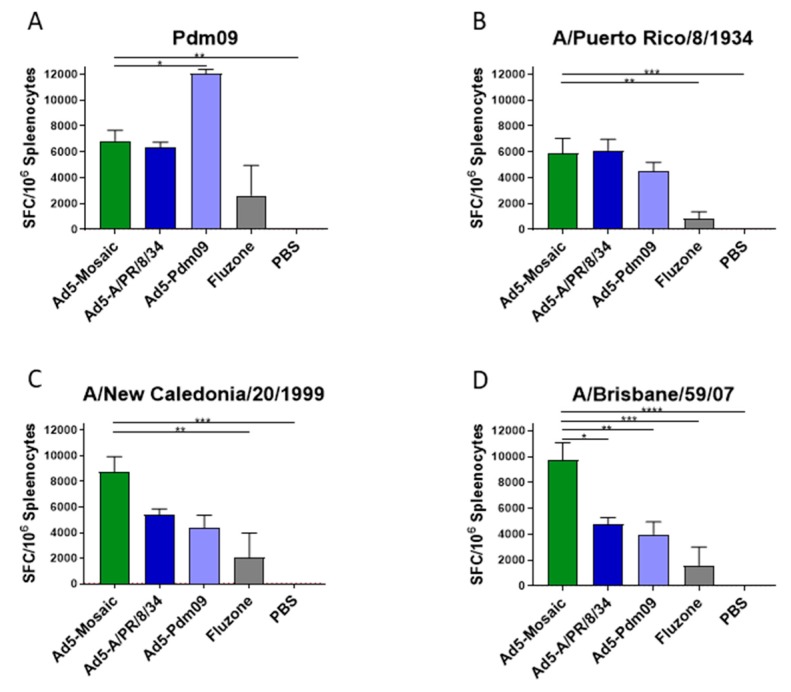
Total T Cell response after single-shot vaccination. Mice were immunized with Ad5-Mosaic, Ad5-A/PR/8/34, Ad5-Pdm09, Fluzone, or PBS and splenocytes were harvested three weeks later (*n* = 8). Total T cell responses were measured by IFN-γ ELISpot assay using peptide arrays spanning the entire HA gene of (**A**) A/California/07/09 (Pdm09), (**B**) A/Puerto Rico/8/34, (**C**) A/New Caledonia/20/99, or (**D**) A/Brisbane/59/07 strains. Data is expressed as the mean spot forming cells (SFC) per million splenocytes with standard error (data was analyzed with one-way ANOVA and Bonferroni multiple comparison and compared to Ad5-Mosaic; **** *p* < 0.0001, *** *p* < 0.001, ** *p* < 0.01, * *p* < 0.05).

**Figure 5 vaccines-07-00195-f005:**
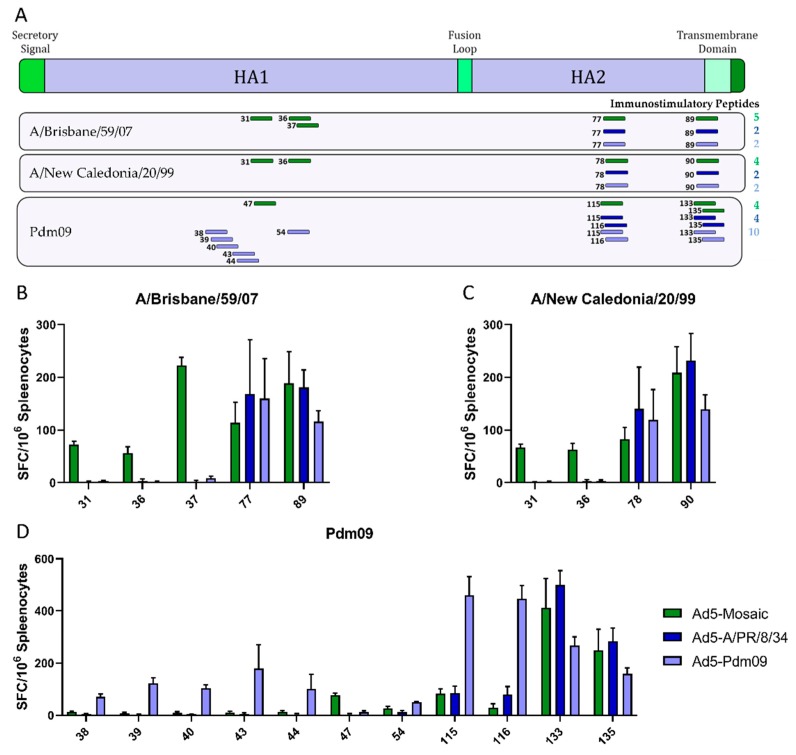
T Cell mapping after single shot vaccination. Mice were immunized with Ad5-Mosaic, Ad5-A/PR/8/34, or Ad5-Pdm09 and splenocytes were harvested three weeks later (*n* = 8). T cell epitopes were mapped using overlapping peptide libraries spanning the entire HA gene. (**A**) Responses greater than 50 spot-forming cells (SFC) per million splenocytes were considered positive and are shown in relation to the HA gene location. Positive peptides for (**B**) A/Brisbane/59/07, (**C**) A/New Caledonia/20/99, and (**D**) A/California/07/09 (Pdm09) are quantified. Numbers correspond to the peptide number in each peptide library. Data is expressed as the mean spot forming cells (SFC) per million splenocytes with standard error.

**Figure 6 vaccines-07-00195-f006:**
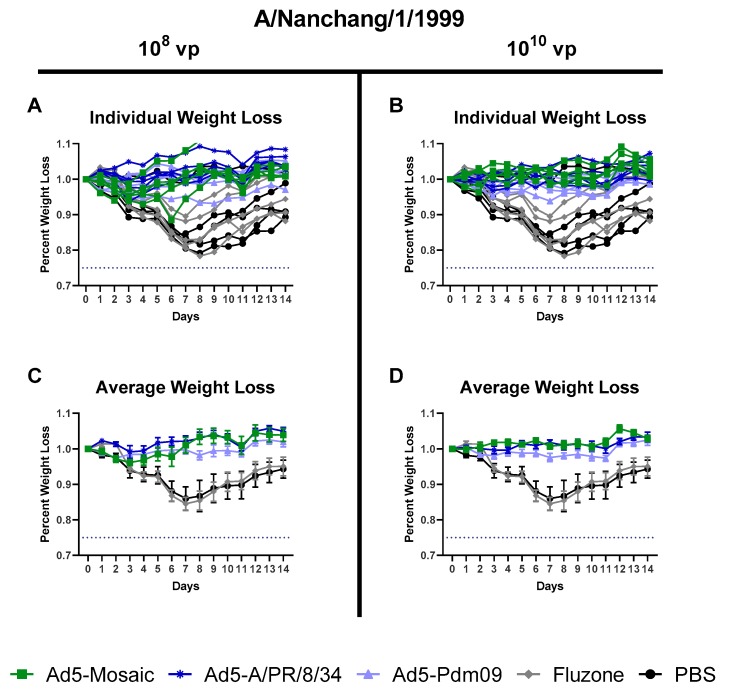
Protection Against A/Nanchang/1/99 sublethal challenge. Mice were immunized with either 10^8^ or 10^10^ virus particles of recombinant Ad5-Mosaic, Ad5-A/PR/8/34, Ad5-Pdm09, 600 ng Fluzone, or PBS and challenged with 10^3.6^ TCID_50_ equivalents of mouse-adapted A/Nanchang/1/99 virus three weeks later (*n* = 5). Mice were monitored daily and humanely euthanized if 25% weight loss was recorded. (**A**,**B**) Weight loss for individual mice and (**C**,**D**) average weight loss are reported (data for average weight loss is reported as the mean with standard error).

**Figure 7 vaccines-07-00195-f007:**
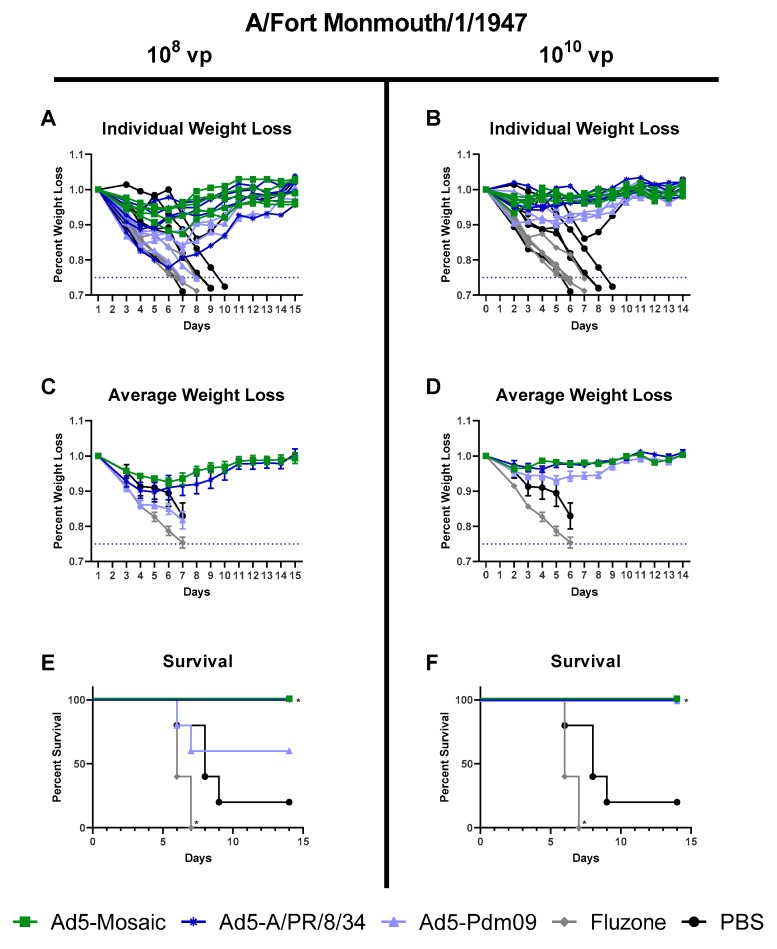
Protection Against A/FM/1/47 lethal challenge. Mice were immunized with either 10^8^ or 10^10^ virus particles of recombinant Ad5-Mosaic, Ad5-A/PR/8/34, Ad5-Pdm09, 600 ng Fluzone, or PBS and challenged with 100MLD_50_ of A/FM/1/47 virus three weeks later (*n* = 5). Mice were monitored daily and humanely euthanized if 25% weight loss was reached. (**A**,**B**) Weight loss for individual mice, (**C**,**D**) average weight loss, and (**E**,**F**) percent survival are reported (data for average weight loss is reported as the mean with standard error; survival data was analyzed with log rank test compared to the PBS control; * *p* < 0.05).

**Figure 8 vaccines-07-00195-f008:**
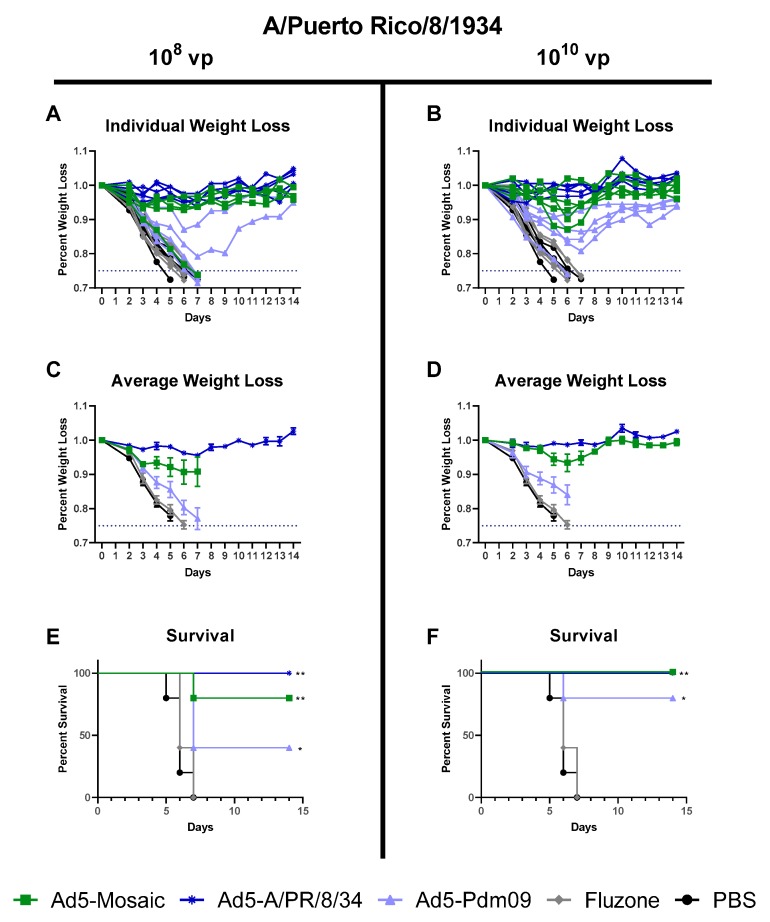
Protection Against A/PR/8/34 lethal challenge. Mice were immunized with either 10^8^ or 10^10^ virus particles of recombinant Ad5-Mosaic, Ad5-A/PR/8/34, Ad5-Pdm09, 600 ng Fluzone, or PBS and challenged with 100MLD_50_ of a A/PR8/34 virus three weeks later (*n* = 5). Mice were monitored daily after challenge and humanely euthanized if 25% weight loss was recorded. (**A**,**B**) Weight loss for individual mice, (**C**,**D**) average weight loss, and (**E**,**F**) percent survival are reported (data for average weight loss is reported as the mean with standard error; survival data was analyzed with log rank test compared to the PBS control; ** *p* < 0.01, * *p* < 0.05).

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
