# Peer review of "Influenza H1 Mosaic Hemagglutinin Vaccine Induces Broad Immunity and Protection in Mice"

_vaccines, 2019, doi:10.3390/vaccines7040195_

Round 1
Reviewer 1 Report
The manuscript by Corder et al. describes production of a novel computer-designed mosaic variant of influenza A virus H1 hemagglutinin and the study of its immunogenicity and protective efficacy in mouse model. This work is scientifically sound and deserved publication. Its main weakness is utilization of a highly immunogenic adenoviral vector, which is unlikely to be used for human vaccination and comparing it with inactivated vaccine. This should be clearly stated in the Discussion when the comparison with Fluzone is being made. It should be also noted that in no protection study presented was this mosaic HA superior to one from A/PR/8/34 strain. Other criticisms are: 1) adequacy of Fluzone dose used (600 ng of HA as stated on p. 5); 2) Confusing captions in Figs 6 through 8, when the designation of viral strain used for challenge is followed by the dose of a recombinant vaccine adenovirus, e.g., in Fig. 6A we see 'A/Nanchang/1/99 10E8 vp', while the dose of A/Nanchang/1/99 used in this experiment is 10E3.6 TCID50 according to Fig legend and 10E8 vp is the dose of recombinant Ad5-HA (but not of Fluzone, which also should be addressed in Fig description).
Two minor comments on language: 1) the last sentence of Intro speaks of 'immune response and protection of monovalent H1 HA...', while it should be 'immune response to' and 'protection induced by, etc.'; 2) Sentence in 3.2 starting with 'Mice immunized with wild type HAs...' should be 'Mice immunized with recombinant Ad5 expressing wild-type HAs, etc.'
Author Response
Dear Reviewers,
Thank you for taking the time to review our manuscript. We have addressed all of your suggestions and critiques. We believe the manuscript is much more complete and significantly improved after revisions. Please see the responses to your comments below. The reviewer comments are in normal text and our response is italicized. You will find that we corrected the manuscript and included the inserted text in the response in order to help you identify improvements.
Reviewer 1:
The manuscript by Corder et al. describes production of a novel computer-designed mosaic variant of influenza A virus H1 hemagglutinin and the study of its immunogenicity and protective efficacy in mouse model. This work is scientifically sound and deserved publication. Its main weakness is utilization of a highly immunogenic adenoviral vector, which is unlikely to be used for human vaccination and comparing it with inactivated vaccine. This should be clearly stated in the Discussion when the comparison with Fluzone is being made.
We agree that there is a dramatic difference in the immunogenicity of Adenovirus and inactivated vaccines. We have addressed this in the discussion as follows:
Discussion, line 437: “Despite increased immunogenicity, adenoviral vectored vaccines are currently not licensed for human vaccine use. Therefore, additional vaccine platforms could be investigated to maximize the immune response to vaccination.”
It should be also noted that in no protection study presented was this mosaic HA superior to one from A/PR/8/34 strain.
This is an important point and we have addressed it as follows:
Discussion, line 442: “For both 100MLD50challenges, Ad5-mosaic vaccinated mice were protected at comparable levels to the matched Ad5-A/PR/8/34 vaccinated animals.”
Section 3.4.2, line 370: “These data show that the mosaic vaccine was able to protect mice against the A/FM/1/47 strain at comparable levels to the similar A/PR/8/34 immunogen. Furthermore, the mosaic vaccine provided better protection than mismatched Pdm09 immunogens or Fluzone.”
Other criticisms are: 1) adequacy of Fluzone dose used (600 ng of HA as stated on p. 5);
We agree that this should be described and have addressed it as follows:
Section 2.9, line 149: “Female BALB/cJ mice were immunized with 1010vp of indicated recombinant Adenovirus 5 (rAd5), 600 ng of Fluzone (approximately 30X a human dose), or control phosphate buffered saline (PBS).”
Section 2.13, line 202: “Groups of 5 female BALB/c mice (~8 weeks old) were immunized intramuscularly at day 0 with either 1010vp or 108vp of Ad5-Mosaic, Ad5-A/PR/8/34, Ad5-Pdm09, control PBS, or 600 ng of HA (approximately 30X a human dose) from the 2018-2019 Fluzone vaccine ...”
Other criticisms are: 2) Confusing captions in Figs 6 through 8, when the designation of viral strain used for challenge is followed by the dose of a recombinant vaccine adenovirus, e.g., in Fig. 6A we see 'A/Nanchang/1/99 10E8 vp', while the dose of A/Nanchang/1/99 used in this experiment is 10E3.6 TCID50 according to Fig legend and 10E8 vp is the dose of recombinant Ad5-HA (but not of Fluzone, which also should be addressed in Fig description).
Agreed. We modified the layout and titles of these figures for clarity. We also added 600 ng Fluzone to each figure legend. See lines 357, 376, and 394.
Two minor comments on language: 1) the last sentence of Intro speaks of 'immune response and protection of monovalent H1 HA...', while it should be 'immune response to' and 'protection induced by, etc.';
The following has been reworded for clarity:
Introduction, line 61: “Here we analyzed the immune response and protection induced by a monovalent H1 HA mosaic immunogen expressed in an adenoviral vector”
Two minor comments on language: 2) Sentence in 3.2 starting with 'Mice immunized with wild type HAs...' should be 'Mice immunized with recombinant Ad5 expressing wild-type HAs, etc.'
The following was reworded for clarity:
Section 3.2, line 257: “Mice immunized with recombinant Ad5 expressing wild type HAs were able to induce robust antibody responses against matched IAV strains.”
Reviewer 2:
Corder et al. have submitted a very interesting and promising Approach for a vaccine candidate against Influenza (H1). Whilst this Topic is broadly addressed worldwide and there is a plethora of literature the Major novel aspect to be highlighted is that their vaccine Approach would protect against a much wider range of H1 variants than other approaches did so far. This is reached by a mosaic hemagglutinin vaccine. Based on the manuscript I have seen I thus recommend to accept the paper.
We thank the reviewer for their time, effort and recommendation of accept.
Sincerely,
Eric A Weaver
Reviewer 2 Report
Corder et al. have submitted a very interesting and promising Approach for a vaccine candidate against Influenza (H1). Whilst this Topic is broadly addressed worldwide and there is a plethora of literature the Major novel aspect to be highlighted is that their vaccine Approach would protect againsts a much wider range of H1 variants than other approaches did so far. This is reached by a mosaic henagglutinin vaccine.
Based on the manuscript I have seen I thus recommend to accept the paper.
Author Response

(The authors gave the same response as above.)
